# Why Temporal Inference Stimulation May Fail in the Human Brain: A Pilot Research Study

**DOI:** 10.3390/biomedicines11071813

**Published:** 2023-06-24

**Authors:** Krisztián Iszak, Simon Mathies Gronemann, Stefanie Meyer, Alexander Hunold, Jana Zschüntzsch, Mathias Bähr, Walter Paulus, Andrea Antal

**Affiliations:** 1Department of Neurology, University Medical Center Göttingen, Georg-August University, 37075 Göttingen, Germanystefanie.meyer@med.uni-goettingen.de (S.M.); j.zschuentzsch@med.uni-goettingen.de (J.Z.); mbaehr@gwdg.de (M.B.); aantal@gwdg.de (A.A.); 2Institute of Biomedical Engineering and Informatics, Ilmenau University of Technology, 98684 Ilmenau, Germany; alexander.hunold@tu-ilmenau.de; 3Department of Neurology, Ludwig-Maximilians University Munich, Klinikum Großhadern, 81377 München, Germany; wpaulus@gwdg.de

**Keywords:** transcranial electrical stimulation, temporal interference stimulation, noninvasive brain stimulation, tES, TIS, NIBS

## Abstract

Temporal interference stimulation (TIS) aims at targeting deep brain areas during transcranial electrical alternating current stimulation (tACS) by generating interference fields at depth. Although its modulatory effects have been demonstrated in animal and human models and stimulation studies, direct experimental evidence is lacking for its utility in humans (in vivo). Herein, we directly test and compare three different structures: firstly, we perform peripheral nerve and muscle stimulation quantifying muscle twitches as readout, secondly, we stimulate peri-orbitally with phosphene perception as a surrogate marker, and thirdly, we attempt to modulate the mean power of alpha oscillations in the occipital area as measured with electroencephalography (EEG). We found strong evidence for stimulation efficacy on the modulated frequency in the PNS, but we found no evidence for its utility in the CNS. Possible reasons for failing to activate CNS targets could be comparatively higher activation thresholds here or inhibitory stimulation components to the carrier frequency interfering with the effects of the modulated signal.

## 1. Introduction

A recent development in transcranial electrical stimulation (tES) strategies is to apply a well-known technique in physiotherapy and acoustics to enhance the depth and focality of the stimulation by employing two channels of sinusoidal frequencies with a slight shift in their frequency values, both in the kHz range to create amplitude-modulated (AM) electric field envelope beating with the difference of the original two currents’ frequencies in an intended area. This strategy, when used for tACS (as in our study), is most often referred to as temporal interference (TI) [1]. Reaching of deeper areas is claimed to be achieved preferentially for the envelope frequency currents because the higher applied frequencies claimed to encounter lower impedance while evade interaction in the more superficial off-target areas due to the supraphysiological nature of the applied kHz stimulation [1,2]. Stimulation protocols with similar interferential design have long been established instruments for electrotherapy and physiotherapy [3], but usually with different waveforms from the continuous sinusoidal ones that are traditionally utilized in transcranial protocols, such as ‘Russian current’, monophasic pulsed currents or other burst-modulated currents [4].

As theorized by Mirzakhalili and colleagues [5], stimulation-induced changes in ion channels, among other effects, will also necessarily rectify the electric potential of the membrane, which leads to imperfect demodulation of the TI stimulus and leads to potential activation or conduction block (CB, described later) in off-target areas, rendering TIS’s clinical implementation restricted.

The basic principle of TI’s mechanism is that when two waves of different frequencies overlap, the amplitude of their sum changes periodically (upon the series of alternating constructive and destructive interference) at the frequency that is equal to the difference of the original two waves’ frequencies. The principle of this interference has been utilized for electrotherapy since the 1950s as interferential current therapy (IFC) and is still used for symptomatic relief and management for chronic intractable pain, and for increasing localized blood flow [6,7]. Typical IFC devices apply one frequency at 4000 Hz and another frequency between 4001 and 4100 Hz, i.e., beat frequencies between 1 and 100 Hz (the difference of the original two current’s frequencies). In comparison to traditional current stimulation (<100 Hz), the kHz frequencies encounter lower skin impedance and can therefore penetrate deeper into the tissues [6,7]. The intended focus can be shifted by adjusting the intensity ratio between the two pairs of stimulating currents [1].

By design, TI stimulation intends to stimulate deep brain areas, where effects are difficult to verify straightforward. In order to disentangle the possible interaction of high-frequency carrier frequencies in the kHz range with the lower TIS target frequencies, we here focus on three different easily analyzable phenomena in human subjects, muscle twitches, retinal phosphenes and EEG alterations.

## 2. Materials and Methods

Our aim is to gain insight into whether TIS can be utilized to deliver amplitude-modulated alternating currents beyond surface-level brain regions, the target of traditional transcranial alternating current stimulation (tACS) protocols, with or without interference of the carrier frequency—more specifically, whether it can bring about the same effects as traditional tACS, but with the added benefit of deeper focus during stimulation. To test this, we first target muscles in the upper limb to gain conceptual proof (‘Experiment 1—muscle study’), then we target the retina to induce phosphene sensation (‘Experiment 2—phosphene study’), and finally, we try to modulate the mean power of alpha waves as measured with EEG (‘Experiment 3—alpha power study’).

### 2.1. Participants

In total, 24 healthy, right-handed, primarily University of Göttingen affiliated adults were recruited from the Göttingen area, Germany through flyers and messaging platforms, snowball sampling the contacts of participants from previous studies. A total of 20 people participated in ‘Experiment 1—muscle study’, and out of these 20 people, 6 also participated in ‘Experiment 2—phosphene study’ and ‘Experiment 3—alpha power study’. Four additional participants were recruited for ‘Experiment 2—phosphene study’ and ‘Experiment 3—alpha power study’ to reach a sample size of ten for these latter two experiments (Appendix B). All participants were between the ages of 19 and 40. Gender ratio was kept equal in all experiments, and none of the participants were pregnant or breastfeeding (for the comprehensive list of exclusion criteria see Appendix C). Participation was first assessed with self-completed questionnaires before the start of the sessions. Compensation for participation was EUR 12 per hour. Subjects were free to terminate their participation at any point in this study without giving any reason. None of the participants were taking chronic or acute medication at the time of this study.

### 2.2. Equipment

Our primary TIS setup was based on a neuroConn TI system: “using two regular ‘DC-STIMULATOR PLUS’ (NeuroConn GmbH, Ilmenau, Germany) with a modified remote circuit board—frequency resolution up to 1 kHz” [8]. The custom modification is a broader bandwidth up to 2 kHz as follows:Two advanced DC Stimulator Plus in Remote control connected to the signal generation unit and a resistor bridge at the output (−3 dB bandwidth > 3 kHz);cDAQ-9136 CompactDAQ controller (running LabVIEW) (National Instruments Corp., Austin, TX, USA);NI9260 dual-channel voltage generation module providing remote control signals (24 bit, 51 ksps) (National Instruments Corp., Austin, TX, USA);NI USB-6255 Multifunction I/O Device with two analog input channels active input range: ±10 V (16 bit 50 ksps) (National Instruments Corp., Austin, TX, USA).

Further descriptions of the customization parameters can be found in the work of Hunold, and Hunold and colleagues [8,9].

The setup was customized in a way that the two current sources were galvanically isolated and temporally synchronized, being driven by signal generators operating on the same clock. The system’s output capability was increased to 4000 μA (peak-to-base; in the following, labelled current strength always refers to peak-to-base). Its function was validated for linearity, current source isolation (no crosstalk between current sources), and the generation of interferences by the Ilmenau laboratory (analysis of 3D measurement arrays in a saline filled aquarium volume conductor. The interference was distinguishable solely across the resistor network), and with spectrum analyzers before and during the studies in our laboratory. This setup made it available for us to change frequencies and intensities in real time, during stimulation. ‘Experiment 2—phosphene study’ and parts of ‘Experiment 1—muscle study’ were also replicated with our secondary setup, an integrated Soterix Medical ‘Interferential Neuromodulation System’ (Soterix Medical Inc., Township, NJ, USA), yielding comparable results.

Electrical stimulation was applied noninvasively via four Technomed reusable surface gold-plated cup electrodes for all three experiments. ‘Experiment 2—phosphene study’ and parts of ‘Experiment 1—muscle study’ were also replicated using 25 mm conductive rubber silver chloride (Ag/AgCl) surface electrodes, again, yielding the same results. In ‘Experiment 1—muscle study’, we used a ‘Philips Affiniti 70′ ultrasound system to determine muscle location, movement isolation, and movement depth. We used the Brainvision Actichamp Plus EEG system for our ‘Experiment 3—alpha power study’.

We used Ten20 conductive electrode paste for the stimulation electrodes in all experiments, and Easycap SuperVisc high-viscosity electrolyte gel for active electrodes under the EEG electrodes during ‘Experiment 3—alpha power study’.

## 3. Experiment 1—Muscle Study

The currents applied here have a continuous sinusoidal shape instead of the pulsating or chopped ones used in Li and colleagues’ study [10]. Whether interferential stimulation interacts with muscle tissue directly or activates the neurons that intervene in the target muscles is unclear. The activation of denervated muscles requires far greater stimulation lengths (up to 100–1000-fold longer duration with pulse stimulation) than the activation of their nerve innervated counterparts [11]. Stimulation of peripheral nerves resulting in motor action of muscles distal of the stimulated nerve would support the latter scenario. In addition to our muscle stimulation targets, we stimulated the ulnar nerve from the distal end of the inner upper arm (approximately 70 mm from the cubital tunnel), close to the medial intermuscular septum to evaluate possible movement of the little finger (digitus minimus manus), which can help shed light on this matter. The upper arm region was chosen in light of the complicated and narrow anatomy of the forearm, disabling the differentiation between direct muscle or nerve action, even if the former assumably require far greater stimulation intensities.

In each condition, the effect of interferential current has been compared with both traditional ACS at the targeted TIS frequency and the kHz carrier frequency stimulation as controls. Our assumption was that we can detect muscle movement on the beat frequencies in the TIS condition, but we cannot observe movement in the carrier control (0 Hz beating) or ACS control conditions. Moreover, we expected that the stimulation’s locus is steerable in the direction of the weaker current.

### 3.1. Methods (Muscle Study)

#### 3.1.1. Stimulation Targets (Muscle Study)

Each participant was stimulated on the left arm in a half-lying position on an inclined bed. The electrode pairs were placed on the skin over the following targets: flexor digitorum superficialis (FDS) muscle, flexor pollicis longus (FPL) muscle, extensor digiti minimi (EDM) muscle, extensor carpi radialis longus (ECRL) muscle, biceps brachii (BB) muscle and the ulnar nerve (UN) which innervates the flexor digiti minimi muscle (FDM; little finger flexor). High precision was not required when targeting these muscles, as any notable muscle movement was detected by the ultrasound system used to verify correct muscle targeting.

The paired electrodes were 45 mm apart (cup edges to cup edges), and the opposing electrodes were 1.5 mm apart from each other (cup edges to cup edges) creating a rectangular arrangement with the motor point (focus point of the stimulation: “where the amplitude of the stimulus required to fully activate the muscle is at minimum” following the definition of [12]) being in the middle of this rectangle. The electrodes were attached via strips of adhesive tape on skin that previously was treated with alcoholic skin cleaner to reduce electrical impedance. To locate the correct position of the targeted muscle, we asked the participant to execute its function (e.g., repeatedly flex the middle finger for finding the FDS) whilst identifying the muscle via ultrasound (Appendix A). Once discovered, the location was marked with a felt tip pen on the skin. The final locations (tip of the horizontal lines) of the cup electrodes were set by determining two points 27.5 mm longitudinally apart from the ultrasound defined muscle location (in both directions) and moving 5.75 mm transversally from these points (Figure 1), yielding four purple dots in total that marked the final locations for the cup electrodes we previously applied with conductive paste.

#### 3.1.2. Stimulation Procedure (Muscle Study)

The maximal current used in this study was 4000 µA (peak-to-base). Stimulation did not take more than 60–70 s per location, and the whole session lasted between 33 and 60 min. Participants were asked to indicate if significant discomfort has been reached while increasing intensity in both channels simultaneously by the same amount.

Stimulation was started at a modulated frequency of 5 Hz at 500 µA intensity that was increased slowly (~1.5 s intervals in the beginning, ~2.5 at the end) by 50 µA increments. Whenever muscle movement was detected, the intensity was noted down and we continued the experiment up to the point where the participant indicated that their pain threshold was reached, at which point we promptly aborted the stimulation. After noting down the pain threshold, we decreased the intensity to 50 Hz above the intensity at their first movement threshold. Now we used ultrasound once again to determine the depth of the muscle movements (in millimeters) and to determine how isolated the muscle movements were on a subjective scale of 1 to 10, where 10 meant perfect isolation and 1 meant that all surrounding muscles were moving too (the rating was done by the same person in each two sessions to increase statistical reliability). The intention of the semiquantitative scoring was to improve distinction of the source of the muscle movement, addressing difficulties of the separation of tissue-movement whether it was due to adjacent muscle co-activation or simply due to their mechanical attachment to the targeted muscle. After rating, we decreased the stimulation delta frequency to 0 Hz (carrier control condition) and held it for five seconds, looking for movements. After the five seconds, we set the Δ frequency to 1 Hz from which we slowly increased it up to the point where the rhythm of muscle movements could not be distinguished into discrete beats anymore and only a continuous, tonic contraction that lasted for more than a second could be observed. The participants then were asked if the stimulation was less comfortable, more comfortable or resulted in the same level of (dis)comfort whenever the stimulation Δ frequency was at either end of the applied frequency range. Finally, in 10 of the 20 participants, we used the acquired pain threshold minus 50 µA to perform conventional ACS (only one active electrode pair delivering currents with 5 Hz, slowly ramped up to 10 Hz) for a final five second as a control condition (ACS control) during which we looked for detectable muscle movements again.

For assessing steerability, each time beat frequency muscle contractions were detected in the FDS muscle, we lowered the current power in one of the electrode pairs at 50 µA decrements. Hypothetically, the beat envelope and the focus point of the TIS is steerable and should move in the direction of the weaker current. FDS was used as a focus point, applying electrode pairs parallel in length with it. Lowering the electrode current placed on the medial side of the forearm should move the focus point into the medial direction and therefore stimulate the FPL instead of the FDS at some point during the intensity decrements.

Anthropometric measurements were taken at the end of the session: wrist circumference (as measured at its smallest radius), forearm circumference (measured at its widest radius), bicep circumference (likewise, measured at their widest radius), forearm length (measured as the distance from the proximal wrist crease to the elbow crease), height (in meters), and weight (in kilograms). BMI was calculated using the latter two variables. In all our results tables, the first 10 participants were recorded by one person, and the second 10 participants by another.

### 3.2. Results (Muscle Study)

The tables below present thresholds (peak-to-base) for contractions (Table 1) and for pain (Table 2).

Based on our results, the most appropriate stimulation site regarding reliability was the FDS as movements were successfully induced in all our subjects but one here (mean threshold: 1872.37 μA, standard deviation: 420.39 μA, Table 1), although one of our participants reported notable discomfort at the intensity which first induced movement (Table 2).

If maximum intensity is a concern (output above 2000 μA), focusing on UN can be advantageous: in our sample, muscle activation when targeting this site occurred reliably at a mean threshold as low as 1158.33 μA (standard deviation: 391.08 μA).

The frequency inducing the largest muscle movement with the lowest threshold was 5 Hz, we could observe movement at this frequency for each successful stimulation attempt. The difference between carrier frequencies (Δ; delta frequency), however, proved to be a very unreliable indicator of discomfort between subjects. Four participants reported notable discomfort when stimulation frequencies were above 5, 7, 7 and 10 Hz, respectively. Higher frequencies decreased stimulation reliability as well (the irregularity of the twitches increased, e.g., in a 10-s stimulation bout with 5 Hz, perfect regularity would mean 50 twitches during the bout. By more irregularity, we mean that there is a larger chance for a twitch to be skipped and only the following to occur: on 5 Hz instead of a 0.2 s break between twitches, a multiplier of 0.2 s passes instead, but they did not alter the movement threshold. The depth and focus of contractions are presented in Appendix E. On-frequency muscle contractions are demonstrated in Appendix A.

During testing for steerability, we could not achieve selective muscle activation, however, some participants noted that the locality of the stimulation sensations moved corresponding to the intended direction of the steering which may indicate subthreshold steering. Due to lower overall intensities (intensity being decreased in one of the electrode pairs), contraction intensity decreased too. We saw that when decreasing intensity in one electrode pair to steer the locus, the net current strength would shift below the movement threshold (it is insufficient to induce muscle movement anymore), so with our output capabilities, we cannot estimate whether the locus of the amplitude maximum moves appropriately. Movement thresholds being too high to adjust ratios meaningfully while keeping the current sums fixed would provoke the question if the failure is muscle specific, or a matter of the distance to the electrode. Although steering with our parameters is not efficient in the PNS, it presumably would be more practical for transcranial targets because of the much smaller distances between cortical structures.

We used Spearman’s rank correlation coefficient to estimate the relationship between motor thresholds and various anthropometric measurements in our sample. The correlation between movement thresholds (averaged through sites, driven mostly by FPL and FDS) and weight (and therefore also body mass index [BMI] to a lesser degree) were detected (N = 20; α = 0.05; Spearman’s ρ = 0.56; *p* = 0.01) (Figure 2). BMI was calculated as the participant’s weight (in kilograms) divided by their height (in meters) squared (kg/m^2^). Note that BMI is not suitable for estimating relative body fat and lean body mass proportions, i.e., measuring obesity.

We report additional observations in Appendix F.

### 3.3. Discussion

First, we confirmed that interferential stimulation does evoke muscle movements with continuous sinusoidal waveform, as utilized by transcranial temporal interference stimulation protocols. Confirming TIS’s utility in the PNS can aid in exploring common peripheral muscle-nerve stimulation techniques to translate into transcranial stimulation paradigms. The translation’s limitations include, however, the significantly larger stimulation amplitude required to evoke muscle movements in the limbs (compared to CNS targets), which might shift the levels of discomfort into being hardly tolerable when applied to target areas that are typical during transcranial stimulation. We found that in our study the most suitable stimulation site was the FDS, using intensities between 2550 and 2850 μA to reliably trigger movement while still minimizing discomfort. Increasing the carrier frequency is also an option to reduce discomfort, albeit there is some evidence in mice that higher carrier frequencies linearly elevate the motor thresholds too when stimulating the motor cortex directly (with a slope of 250 µA/kHz) [1]. Stimulating the UN can be advantageous if the output capabilities of stimulators are limited as this site requires only approximately 1000 μA intensity on average for successful stimulation; however, choosing one of these three sites might surrender some reliability of movement induction, as we failed to induce movement in some of our participants at these sites. We had two unsuccessful attempts at stimulating ECRL due to the strong co-activation of surrounding tissues that made determining if movement was induced in the ECRL unfeasible. This result is concerning for the possibility of reaching depth with TIS and needs further investigation. Our most appropriate stimulation frequency was 5 Hz and below, and anything above 7 Hz is to be avoided due to more discomfort for some participants without seemingly any advantages in return. This is possibly because of tetanic stimulation building up a continuous contraction [13]. The rate of observed muscle movements always followed the stimulation Δ frequency or its subharmonics up to approximately 12 Hz, except for some cases during biceps stimulation when contractions sometimes were sustained during stimulation (but still painless). The intensity of the twitches seemed not to fatigue up to a few minutes of stimulation, but we did not continue to test for endurance. This seems to be roughly in line with the observations of Bowman and McNeal’s conduction block experiment [14] in feline sciatic nerve. The amplitude modulation may produce a beating that is imperfect but very close to the difference in frequencies (below 1 Hz mean ± SD on trials with 10 Hz Δ frequency) [1].

We also found that higher bodyweight correlates with higher movement thresholds. It is known that increased electrode sizes and/or increases in inter-electrode distance elevate thresholds, but conclusions about the relationship of body weight/adiposity and thresholds are indecisive, correlations in both directions have been shown [15,16,17]. Participants were not examined for body composition, and correlations between anthropometric measurements and thresholds were sporadic so we cannot draw conclusions on these fronts.

The core question arises whether contractions are induced by direct muscle activation or the activation of the intervening efferent motor nerve fibers that synaptically activate the muscle fibers. Physiologically nerve fibers innervate the muscle cells at the motor point. Without this endplate apparatus and the signal amplifying effects of the nerve-muscle synapse, the electrical signal has to be substantially heightened to achieve whole muscle contractions [12], with a threshold increased an order of magnitude higher than required for nerve activation in cats [18], or in human model [19]. If correct, then nerve action will appear long before muscle activation with the latter requiring intensities that potentially can damage nearby tissue [19].

Nerve targeting is also supported by peripheral stimulation practices, such as in Burrell and coworker’s study [20], which brought evidence for regenerative effects of electrical stimulation in porcine model, by Jabban and colleague’s work [21] on porcine UN stimulation, and by the study of Botzanowski and colleagues [22], validating TI in murine sciatic nerve. The mechanism of indirect stimulation was demonstrated in our study during the stimulation of the UN, when the locus of stimulation was far upstream from the contracting muscle tissue. Although we did not measure angles of contraction, the recruitment curve (threshold to maximal movement) is generally much steeper for nervous action than for muscular activation [18]. This effect is amplified the closer the motor point is to the nerve entry point. Abrupt changes in movement complexity and muscles engaged during stimulation may indicate the shifting of motor point during contraction.

Unfortunately, movement thresholds were too high to adjust ratios meaningfully without drastically decreasing current intensity sums, making steering unfeasible with intensities up to 4 mA (peak-to-base). It presumably would be more practical when targeting the brain directly because of the much smaller distances between cortical structures.

Targets that did not evoke any movement even after throughout examinations were the whole length of the hamstring muscle, the tibialis anterior and surrounding calf muscles, the distal fibers of the quadriceps femoris muscles, the gluteus medius, all the deltoid muscles (when targeted directly), and all the triceps brachii muscles.

In the case of stimulation of the peroneal nerve near the popliteal fossa, two participants described a characteristic pre-movement feeling which can be felt also in the upper limb during stimulation when the applied intensities are slightly below movement threshold.

Further anthropometric parameters worthy of exploration could be the level of hypertrophy of the musculature or body fat percentage (adiposity), although there are difficulties with the objective measurement of these properties [23]. A potential clinical application of TIS is using it as an additional diagnostic tool for neuromuscular diseases, e.g., measuring fatigue in myasthenia gravis.

## 4. Experiment 2—Phosphene Study

Both the stimulation of the retina and the visual cortex can evoke visual perceptions called phosphenes. Exciting the retina with alternating currents evoke strobe light-like, diffuse, flickering flashes of light on a rate that is related to the stimulating current’s frequency [24,25]. The location of the phosphenes in the visual field correlates with the amplitude peaks of the current density distribution on the retina [26].

The data for this experiment were recorded during a brief session held immediately after ‘Experiment 3—alpha power study’ and took no longer than 10 min. Our assumption was that participants will experience phosphenes in both the ACS control and the TIS conditions, but not in the carrier control condition.

### 4.1. Methods (Phosphene Study)

#### 4.1.1. Experimental Procedure (Phosphene Study)

‘Experiment 2—phosphene study’ study followed ‘Experiment 3—alpha power study’ immediately after the second EEG session.

We arranged the electrodes in a rectangular fashion with them being 38 mm apart laterally and 25 mm apart vertically, the pupil of the right eye being in the center. The upper two electrodes were attached to the skin between the upper eyelid and the eyebrow, and the lower ones immediately below the lower eyelid crease (Figure 3). The electrode cables were arranged so that they lay vertically on the face.

#### 4.1.2. Control Stimulation

Before proceeding with the TI stimulation, we used conventional ACS as a control condition in order to train participants. All conditions were performed first with the lights on in the room, and then with the lights turned off immediately after. Only the two lower electrodes were turned on, delivering horizontal currents with a frequency of 15 Hz, and with an initial current strength of 50 μA, and 100 μA already elicited phosphenes for all 10 participants in this study. Phosphene sensations are immediate and obvious, so stimulations took no longer than a few seconds. All participants experienced phosphenes during the ACS control stimulation to establish the general sensation at the lowest used current strength of 100 μA.

#### 4.1.3. TI Stimulation

With phosphene sensations established, we tried to trigger the same sensation with the same electrode setup, but in a temporally interfering manner: both electrode pairs were turned on, one pair delivering 2000 Hz frequency AC while the other one delivering 2010 Hz AC with 100 μA initial intensity that was slowly increased until the participant either sensed phosphenes or indicated in any manner that a considerable level of discomfort was reached. Finally, we set both electrode pairs to 2000 Hz frequency, adjusting the frequency difference to 0 Hz between currents, applying another control condition (carrier control condition).

### 4.2. Results

No participant experienced any phosphene sensation during TIS (up to 4000 µA) or the carrier control condition in either illumination condition. Phosphene sensations only returned once the intensity was tuned below 100 Hz which does not imply interference and can be attributed to the regular ACS phosphene induction.

### 4.3. Exploratory Sessions

Phosphene induction is a very robust and reliable marker of successful photoreceptor stimulation with reasonably uniform between-subjects outcome, so much so that inducting it unintentionally can be a major concern during the design of ACS protocols concerning cognitive performance [27].

We tried the following two configurations in four participants upon setting up our original TI arrangement in the hope of coming up with further ideas: our first setup was the same as in our primary experiment, but the electrodes were plugged in paired vertically and then diagonally instead of horizontally (Figure 3). Our second setup was identical, except that we let participants move the electrodes and/or the position of their cables themselves as they wished during the stimulation to find a possible hotspot for phosphene induction. Another attempt was to move the two electrodes which were proximal to the nose to the other side of the nasal bridge, while correspondingly moving the other pair further away from the nose (distally), so that the focus point remained in the retinal area. Finally, we targeted the visual cortex transcranially from the occipital scalp region. The montages used are shown in Figure 4. Note that no EEG recording has taken place, the sites were used for stimulating (output) electrodes only. In these cases, the electrodes were paired in every possible way too (vertically, horizontally, and diagonally). The electrode cables were first positioned to rest vertically in both montages then their position was gently manipulated online too.

As in our primary experiment, none of these approaches produced phosphene sensations except for the ACS control conditions, even at intensities up to 4000 μA.

High enough intensities (>2300 μA) used during transocular TIS caused muscle twitching around the eyes, reinforcing the success of TIS in muscle (Appendix A). Muscle movements near the movement threshold were described as curious or uncanny, but not as painful.

### 4.4. Discussion (Phosphene Study)

Stimulation intensities up to 4000 μA in the TIS condition produced no phosphenes, they were reported only in the ACS condition. It is possible that even higher intensities are required to induce them, but the fact that muscles responded to much lower intensities during peripheral stimulation, and the comparatively very low required intensity in the ACS condition leads us to believe that there might be other phenomena in play that prevents neural excitement. Furthermore, previous simulation studies in humans suggest that lower intensities should be sufficient [26,27]. One possible explanation is the so called “conduction block” (CB) [4,5], which is a reversible neural block inducted by high-frequency alternating currents, modelled by Wedensky in 1903 [28], a result of the “Gildemeister effect” in which the subsequent de- and repolarization of cell membranes upon ACS exposure lead to a temporal summation effect [29]. The existence of CB later was directly observed by Tanner in 1962 using 20 kHz AC to block the propagation of frog sciatic nerve fiber signals [30]. In our experiment, we operated with frequencies far from the scale of 20 kHz, but later studies done by Bowman and McNeal on feline sciatic nerves [14] confirmed the presence of CB induced by pulse trains even at the same frequency that we used (2 kHz): during a 10 s timeframe, the neural firing around the target area decreased from 1 kHz to 0 Hz, indicating the presence of a sustained and complete CB during stimulation (higher intensity stimulation resulted in faster CB, and recovery was complete by one second after the cessation of the stimulation) [5]. Based on these results there seems to be an optimal range of carrier frequencies where the lowest frequency is still high enough so that physiological interactions are avoided, but low enough to circumvent (or at least delay) the set in of CB. More recent studies suggest that while CB contributes to decreased neural communication, tonic activity modulation [5] and oscillation modulation [9,31] are also contributing factors.

To test for CB induction in our setting, another 2 kHz signal was instituted on three participants at the same time of the traditional phosphene provoking ACS. For the intervening high-frequency signal, we tried using both ACS (one electrode pair) and TIS (two electrode pairs) stimulations, in the latter case having two 2000 Hz outputs in one condition, and one 2000 Hz and one 2015 Hz (15 Hz Δ frequency) outputs in another. All other experimental arrangements were identical to the original experiment. We could not block phosphene sensations with the additional high-frequency stimulation currents and therefore could not find evidence for CB being present during high-frequency retinal stimulation.

## 5. Experiment 3—Modulation of EEG Oscillations Using Alpha Frequencies

Several studies have shown tACS to influence alpha frequency oscillations [32,33,34] as well as the presence of the modulatory effect long after the end of the stimulation, from a minimum of 30 min up to 70 min [35]. Most of these studies target the occipital or occipito-parietal regions. What drives the alpha power modulation remains unclear. Changes have been explained classically by two possible mechanisms: by the entrainment (temporal alignment) of exogenous oscillations with the endogenous brain oscillations, and by lasting plasticity induced by tACS. Vossen and colleagues’ [34] findings (which we based our experimental procedure on) support the latter explanation, or at least that plasticity is the primary driver of modulation. In their study, alpha enhancement was achieved regardless of the stimulation being phase continuous or phase discontinuous to the endogenous oscillations and the enhancement was observed most prominently at the spontaneous alpha peak frequency of participants. Furthermore, delivering frequencies that were tuned in a way to match the intrinsic alpha frequency seemed not to strengthen the enhancement. The authors found no evidence for phase locking either in between stimulation bursts, or immediately after intervention, ruling out phase locking as a driver of sustained alpha modulation, a phenomenon that is typically observable long after the cessation of stimulation. For these reasons, we did not manipulate the phase of our stimulation output or set its frequency on an individual basis. Omitting the latter setting was motivated by numerous other past studies too [32].

The literature is somewhat divided along the lines of comparing either maximum alpha amplitudes or mean alpha powers (the strength of the synchronized brain activity in the alpha range). We decided to use the latter measurement as it seems to be more robust to noise, especially considering the difference in the number of segments we gained after the segmentation of our data [36,37,38]. Alpha range is mostly put between 7.5 and 12.5 Hz or 8 and 12 Hz. We are using the former convention. This activity is most pronounced in the occipital regions, so our chosen electrode sites were C3-O1, and C4-O2 in pairs.

We employed four types of stimulation conditions: TIS, TACS, CARRIER, and SHAM (conditions are detailed in the following section). We hypothesized that stimulation would cause a change in the power spectrum of occipital brain oscillations in both the TACS and the TIS conditions, but not in the CARRIER and SHAM conditions.

### 5.1. Methods (Alpha Power Study)

#### Experimental Procedure

Impedances were always kept below 22 kΩ and usually went below 6 kΩ by the time of the first active stimulation phase. EEG was recorded from 27 sites with a 32 channel EasyCap EEG recording cap (Brain Products GmbH, Gilching, Germany). The five inactive sites were C3, C4, O1 and O2 for the stimulating electrodes and Cz for the reference electrode. We recorded electrical activity along the scalp for four minutes both before and after the stimulation, interrupted by a 5 min stimulation period. Both four-minute recording intervals gathered data with the participants having their eyes open for the first two minutes and having them closed for the latter two minutes. Maximum current intensity was 4 mA, but the exact values were acquired at the beginning of the first stimulation session individually with a staircase procedure by asking participants to notify experimenters when a point of notable discomfort was reached, disabling them from comfortably undergoing 5 min of stimulation at that intensity level. At this point, we decreased the stimulation intensity by 50 μA. Discomfort has invariably decreased to a significant degree with time and so participants got habituated to the stimulation without being distracted by the currents after the first 8–10 s.

We had three active and one sham conditions in a within-subject design:The TIS condition was performed using a O1-C3 and O2-C4 electrode montage and a frequency pair of 2000 Hz and 2010 Hz resulting in a 10 Hz beat frequency.For the *CARRIER* control condition, we used the same montage but with both electrode pairs delivering 2000 Hz frequency resulting in no beat frequency.The TACS condition was performed using a O1-C4 and O2-C3 montage (as opposed to the O1-C3 and O2-C4 ones that we used in our other conditions), creating a diagonal arrangement in which only the O2-C3 pair was active during open eye tACS, and only the O1-C4 pair during closed eye tACS.Our SHAM control condition was performed in the interference montage arrangement (2 pairs of active electrodes), and it involved a ramp up period of 25 s to the intensity previously acquired during our TACS condition, and then back to 0 μA at the start and at the end of our 5 min stimulation period. No active stimulation was performed outside of these 50 s.

For the interferential conditions, we decided in favor of a wider montage, mirroring the setup of Grossman et al. [1], but for tACS we kept a more traditional one for better focus seemingly without losing anything in return.

The stimulation intensities used in the TACS and SHAM conditions were between 1700 μA and 3300 μA, and in TIS and CARRIER conditions between 3300 μA and 4000 μA. The averages were 2500 μA for TACS and SHAM and 3450 μA for TIS and CARRIER conditions.

There was a minimum of 7 days interval between the two sessions. The experiment was conducted in a pseudo-randomized crossover fashion (Table 3).

Participants were randomly divided up into two groups, and the sessions were counterbalanced:

‘Group 1’ started with Session 1 on their first day and Session 2 on their second, while ‘Group 2’ had it the other way: Session 2 on their first day and Session 1 on their first one.

The order of applied conditions in Session 1 was TI (open eyes) -> TI (closed eyes) -> CARRIER (open eyes) -> CARRIER (closed eyes). For Session 2, it was tACS (open eyes) -> tACS (closed eyes) -> SHAM (open eyes) -> SHAM (closed eyes).

There is an inverse relationship between the frequency of tACS and experienced discomfort in general [39]. This has allowed us to tune intensities significantly higher during our interference conditions (TIS and CARRIER) than in the TACS condition without causing additional discomfort. To keep the discomfort levels even lower, we applied a Lidocaine gel containing 2% xylocaine below the stimulating electrodes before the start of the experiment as well as between the second and third stimulation periods. This has in turn allowed us to approach intensities in the TACS condition that we used during the TIS and CARRIER conditions.

The preprocessing steps performed are detailed in Appendix G.

### 5.2. Results (Alpha Power Study)

EEG recording data used for comparisons were from electrode sites Cp1, Cp2, Oz, P3, P4, and Pz. To compare normalized relative changes (dB), we calculated change values, again following the convention of Vossen et al. [34]: (change = 10 ∗ log10 [post-test/pre-test]) and compared them between the elements of our CONDITION variable (SHAM, CARRIER, TI, TACS). For comparison, we used Friedman’s test.

Elevated alpha powers were observed only after the tACS intervention in our sample (Table 4), an effect already established in literature [34]. Alpha powers shrunk in each other condition by roughly the same amount. Testing for significance, the Friedman test (N = 10; α = 0.05) suggests the effect of the *CONDITION* variable (χ^2^(3) = 33.560; *p* < 0.001), with changes being significant between SHAM and TACS conditions (W = 244, *p* < 0.001), and CARRIER and TACS conditions (W = 202, *p* < 0.001) into the positive direction as per two-sided Wilcoxon Rank Sum tests (α = 0.05). Changes between TIS and any other conditions were not significant.

All changes were positive in the TACS condition, and all were negative for all other conditions concerning the six compared electrode sites (Figure 5, Table 5).

### 5.3. Discussion (Alpha Power Study)

tACS leading to elevated alpha powers is in line with the results of previous studies, and our enhancement rates are most similar too [34]. TIS has failed to increase the EEG power which could partially be attributed to the low sample size; however, the TACS condition did lead to significant alpha power changes at a sample size which was similar to or lower than what was typically used in previous studies [34,40], and the values not only not increased, but shifted into the opposite direction, following a similar effect size than the control conditions, so it is unlikely that both TIS and the control conditions had a phasic effect. The values of both the CARRIER control and TIS could speak towards the set in of CB, but the negative shift was even more pronounced for SHAM. Values decreased throughout the whole cortex so if we still suspect the onset of CB, its effect should be considered immense and consequential.

Alpha power modulation is a relatively successful protocol, but studies with our electrode montage (C3-O1 and C4-O2) are less numerous and it is possible that distance is more of a concern regarding electrode positions in interferential stimulations as opposed to traditional ones. The null finding is interesting considering TIS’s success in case of peripheral stimulation. The clear decrease in alpha-power must be noted also. We cannot exclude that the (control) tACS disturbed the intrinsic alpha signal generating neural network, or if this decrease can adequately be explained by the prolonged idling of participants during the experiment.

## 6. Conclusions

We found evidence for TIS utility in the PNS, but not in the CNS. Presently, intrinsic alpha power generation and its interaction with exogenous electric signal is not well understood [41], but the lack of reported phosphene sensations is still puzzling given the stimulation’s success in the PNS. Another concern is the occasional co-activation of adjacent muscles to the intended one during peripheral stimulation. Decreased topographical specificity can obstruct current penetration depth, which further emphasizes the need for designing stimulation montages to enhance stimulation separation at the biological object.

Successful focal modulation in the depth of a well-defined, low-frequency electric field envelope in the muscles of the upper limb was found to be a reliable technique. There is no apparent reason to doubt the electric field formation elsewhere in the body, such as the retina, which makes the unsuccessful phosphene induction surprising considering the free movement of surface electrodes, and the extraordinary (in case of retinal simulation) intensities used. It should be noted, however, that brain tissue is much more anisotropic than that in the limbs. According to our experiments, CB did not explain the lack of nerve activation.

In Mirzakhalili and colleagues’ model [5], passive axons (axons without ion channels) did not contribute to the facilitation of TIS, only active axons (axons with ion channels) did as the Na^+^ ion channel’s conductance was greater relative to the K^+^ channels conductance which created net inward currents, leading to depolarization. Further, the widths of IPSPs and EPSPs are shorter in inhibitory cells than in excitatory cells as summarized in Paulus and Rothwell [42], which may facilitate inhibition at high frequencies in structures where simultaneous inhibition and excitation is present.

In another influential study, Vossen and colleagues [34] argued that spike-timing dependent plasticity (STDP; the hypothesis that the synaptic strength modulation between neurons depends on the order and timing of synaptic potentials) is the primary driver after testing for plasticity by measuring the strength of the aftereffects with regard to phase continuity between a series of tACS bursts. They also investigated how much the stimulation frequency matched the intrinsic alpha frequency post-intervention as a means of testing for entrainment. They found no evidence for entrainment and concluded that plasticity mechanisms offer sufficient explanation for the modulatory effects of tACS. Their results influenced our decisions while designing the stimulation parameters of our third experiment (‘Experiment 3—alpha power modulation’). In the study of Alekseichuk and colleagues [43], the authors found that while anodal tDCS over the occipital cortex slightly increased the online visual task-related BOLD response without observable offline effects, 10 Hz tACS has shown no online effect, but it did show a diffuse (ranging through occipital, temporal and frontal areas) decrease in the BOLD responses elicited by the visual stimulus offline, implying that both tES techniques have effects on neuronal metabolism, although via different mechanisms of action. The latter finding was replicated by Vosskuhl and colleagues [44] too, yielding the same visual task-related BOLD response decrease to alpha frequency tACS, but mostly at areas where the BOLD signal correlated negatively with the alpha amplitude, and without direct tACS effect on resting state BOLD responses.

The precise characteristics of spatiotemporal distribution intracranially remain unclear. In all forms of tES, the applied current is relatively weak (typically below 3 mA), and only about half of that penetrates intracranially, but it still proves to be strong enough to change the probability of action potentials happening in the target areas [45].

Opitz and colleagues’ [46] stereotactic alpha modulation in cebus monkeys and surgical epilepsy patients revealed a small frequency dependent magnitude decrease and phase shift due to tES. Their findings reflected the superficial nature of traditional tES with maximum densities of 0.5 mV/mm. These results suggest modulation by currents with only miniscule strengths actively engaging neurons. In Mirzakhalili and team’s study [5], the TI currents’ propagation in their fine-grained finite element modeling has shown very limited intracranial reach because of the rapid power dissipation upon penetration.

Esmaeilpour and colleagues [30] have shown that AM electric fields have their highest strength at distance in the deeper brain regions while unmodulated electric fields are the strongest at more superficial (cortical) areas; therefore, the neural interaction with the phasic modulation at Δ frequency only happens at the target area. The authors attributed the modulatory exclusion of superficial areas to the intrinsic low-pass filtering properties of the system in contrast to Mirzakhalili and colleagues’ position. The authors have also shown GABAergic (γ-aminobutyric acid; GABA, specifically: GABA_B_) adaptation (plasticity) and its elevating effect on AM waveform sensitivity, especially in the gamma-frequency range. At the crux of this nonlinearity is the stimulation parameters adjustment eliciting different rates of responses on the transmitter-receptor level. The long-term plastic effects of tES are thought to depend on glutaminergic mechanisms, a function of the receptors of calcium, N-methyl-D-aspartate (NMDA), and α-amino-3-hydroxy-5-methyl-4-isoxazoleprionic acid (aminomethylphosphonic acid; AMPA) [47]. While the after-effects are thought to be NMDA mediated, the inhibitory effects are associated with GABA reduction which contraindicates tES in GABA depleted states such as veisalgia (alcohol hangover). GABA**_A_** receptor agonist (Iorazepam) administration is known to enhance a delayed onset excitability elevation induced by tDCS [48].

Mirzakhalili and colleagues [5] came up with suggestions to test for the CB in simplified models with careful controls to aid avoiding the unintentional consequences of off-target CB induction. They argued that high- and low-frequency components of the current are multiplied and not summed by each other based on Oppenheim, Schafer and Stockham’s [49] previous examination of the nonlinearity of signal filtering where the signals are the products or components of a generalized superposition model, theoretically applicable for the output of our TI generating machine too.

Further reach and selectivity were achieved by Lee and colleagues [50] by implanting the electrodes into the epidural space of the skull (rendering it minimally invasive instead of the noninvasive type).

The accurate simulation of tES is achievable only by the fine-tuning of models with more precise conductivity indices about the different tissue types and their structure in the human head are acquired, including peri- and extracranial components such as the layers of skin with its perspiratory properties. Until we attain near-perfect comprehension, we rely on approximate solutions that are immensely challenging to apply due to mere mathematical difficulties in solving for differential equations (Maxwell’s equations) in anisotropic media [5,31,51]. The most popular simulation methods are using simplified models based on one dimension cable theory to calculate the local field values for different neuronal (and sometimes glial) compartments. Developing analytical (exact) solutions to problems applicable only in very specific cases, yielding hybrid analytical/numerical solutions as an end result. Interindividual variances in tissue composition and hence differences in electric conduction are only approximated and presently incalculable to a sufficient precision. Accounting for these variances in the conductivity of cellular tissue and extracellular media, and differences in neural morphology and topography seem to be the biggest challenges during designing stimulations. Cell orientation already plays a significant role in traditional electrical stimulation techniques, possibly even more so during TIS. Arlotti and colleagues [52] tested how field direction and neuronal morphology affects the neuronal excitability with uniform steady-state electric fields. One significant parameter is diameter: the response of cells is frequency-coded, it can respond to a wide spectrum of frequencies, but their activation threshold is always the lowest in a given limited range [5,52]. On top of somatic differences, several modelling results display significant variance in the polarization properties of different axonal and dendritic compartments. As emphasized previously, polarization angle is another significant variant in stimulation success. The main axonal branch laying tangentially or radially (parallel and normal to the cortical surface, respectively) to the electromagnetic field will lead to better conduction along the axon and so optimal overall terminal polarization of the compartment as presented by Mirzakhalili and colleagues [5]. In their study, the probability of AP generation in the terminal compartment were the highest near the nodes of Ranvier, and this probability maximum corresponded to the electric field’s amplitude modulation’s maximum, while at longitudinal axons, action potentials were most frequently evoked in line with the activating functions’ maximum amplitude modulation (the second space derivative of the extracellular medium [53]). With this, the former mechanism is thought to contribute more to the net AP output of the system at terminal parts, and the latter is seemingly more involved in AP generation at axonal sites [5,53].

This work is an exploratory, proof-of-concept study; however, possible clinical utilization of this technique is the above-mentioned aid during the diagnosis of myasthenia gravis. techniques to reduce discomfort or reaching deeper muscle tissue with comparatively lower applied current strength, a desirable goal in disciples such as physical electrotherapy. A follow-up study is already on its way involving up to 80 patients with different neuropathic or myopathic diseases, neuromuscular junction disorders and healthy controls to evaluate differences in subjects’ response to TIS in light of alterations of the neuromuscular unit. Next to elucidating disease-specific response patterns, insights into peripheral TIS and muscle compartment stimulation can shed light on matters that are unfeasible to test for in the CNS and therefore can be powerful tools supplementing transcranial protocols. These insights can improve our understanding of the TIS mechanism itself, allowing for translation to higher complexity levels.

## Figures and Tables

**Figure 1 biomedicines-11-01813-f001:**
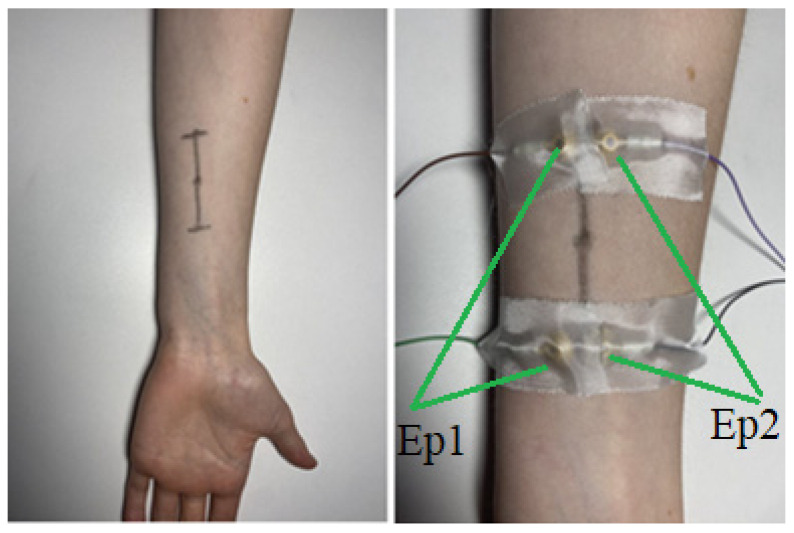
Arranging the electrodes in a long rectangle allows for focused stimulation while still permits us applying the transducer. Ep1: first electrode pair; Ep2: second electrode pair.

**Figure 2 biomedicines-11-01813-f002:**
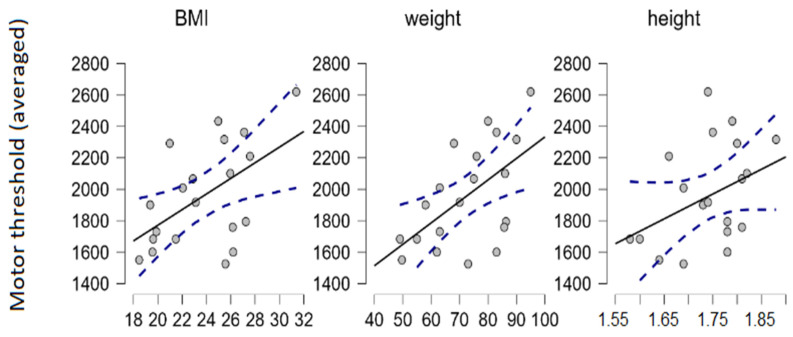
Correlations between motor threshold (averaged through all stimulation sites, in μA) and BMI, motor threshold and weight (in kilograms), and motor threshold and height (in meters).

**Figure 3 biomedicines-11-01813-f003:**
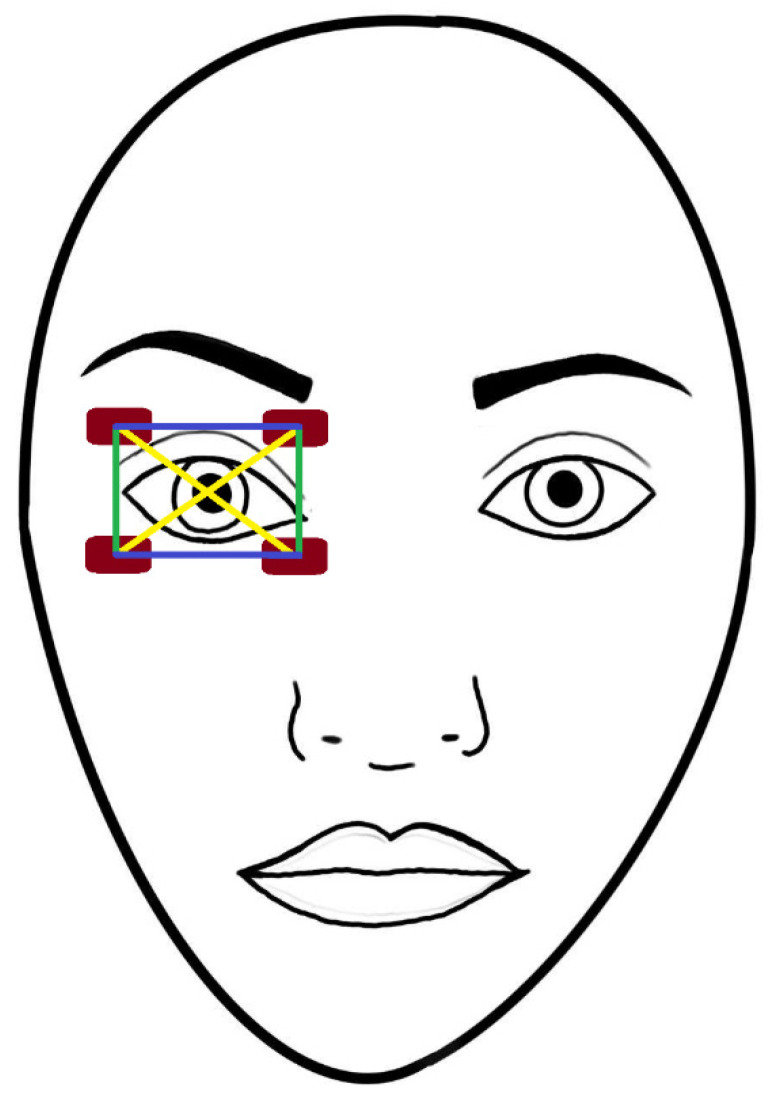
Setup for transorbital TIS of the retina. In our primary study, we stimulated the participants with the electrodes being in pair horizontally (blue lines), but in our exploratory sessions, we also tried vertical (green lines) and diagonal (yellow lines) setups. In the control stimulation, only the two lower electrodes were turned on.

**Figure 4 biomedicines-11-01813-f004:**
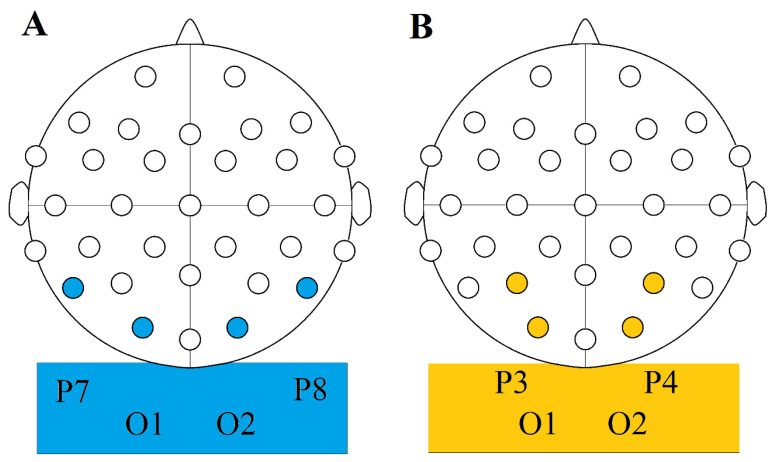
The two electrode arrangements aiming to induce cortical phosphenes directly. (**A**) one montage aimed roughly at the O1, O2, P7 and P8 electrode sites of the international 10/20 EEG system (used throughout the whole study). (**B**) another montage targeting the O1, O2, P3, and P4 electrode sites of the same system.

**Figure 5 biomedicines-11-01813-f005:**
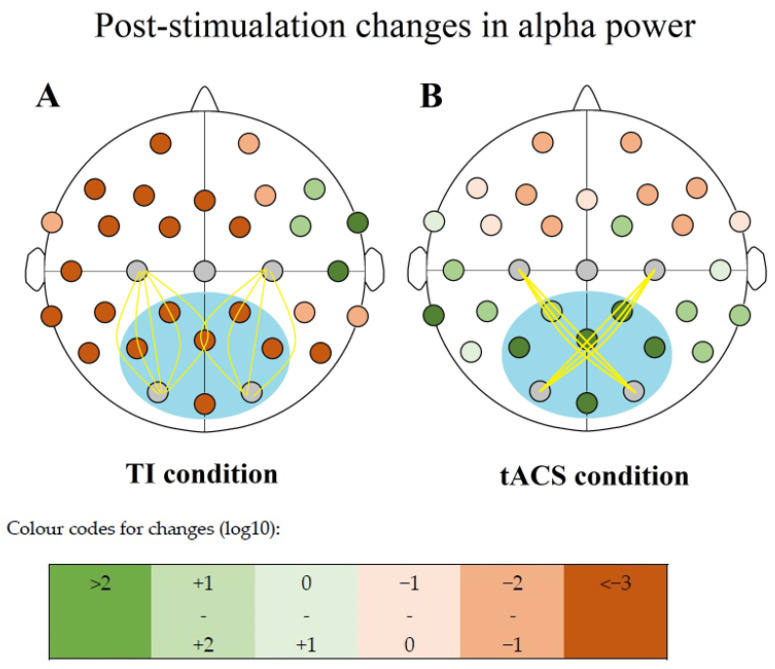
Changes in alpha power upon two conditions represented by the colors explained by the chart above. Inside the blue circle are the six recording electrode sites used for the measurement of changes: Cp1, Cp2, Oz, P3, P4, and Pz. (**A**) exhibits O1-C3 and O2-C4 stimulation sites being in pair, while (**B**) shows O1-C4 and O2-C3 as pairs.

**Table 1 biomedicines-11-01813-t001:** Thresholds for muscle contractions.

	FPL	FDS	EDM	ECRL	BB	UN	x̅′:
P1 (f)	-	1600	-	-	-	1600	1600.00
P2 (f)	1700	1150	-	2000	2200	700	1550.00
P3 (f)	1700	1675	1900	1900	900	1075	1525.00
P4 (m)	2300	1800	-	2200	-	875	1793.75
P5 (f)	2300	2200	-	2500	2750	1300	2210.00
P6 (m)	3100	2550	2700	-	1100	-	2362.50
P7 (m)	1500	2100	2100	2450	-	500	1730.00
P8 (f)	1900	1700	2100	2100	1600	700	1683.33
P9 (m)	-	1000	2000	2300	1100	-	1600.00
P10 (m)	1850	2000	1800	1900	1700	1300	1758.33
P11 (f)	1800	1700	1800	2300	1800	700	1683.33
P12 (f)	1600	1500	2200	2900	2000	1200	1900.00
P13 (f)	2800	-	2900	3300	3000	1100	2620.00
P14 (f)	2800	1900	2400	2850	2100	1700	2291.67
P15 (m)	2700	1700	1900	2900	1500	1700	2066.67
P16 (m)	2300	2000	2300	2700	1500	1800	2100.00
P17 (f)	2600	1800	2100	2650	2000	900	2008.33
P18 (m)	2700	2500	3200	2500	2200	1500	2433.33
P19 (m)	1800	2500	2600	1800	1800	1000	1916.67
P20 (m)	2800	2200	2700	3000	2000	1200	2316.67
x̅:	2236.111	1872.368	2293.75	2458.333	1838.235	1158.333	
σ:	511	420.3913	418.6785	430.8849	551.5686	391.0769	

All values are in μA. FPL: flexor pollicis longus; FDS: flexor digitorum superficialis; EDM: extensor digiti minimi; ECRL: extensor carpi radialis longus; BB: biceps brachii; UN: ulnar nerve. Missing data points mean we could not induce movement at a given location. Neither the carrier nor the ACS control conditions produced any movements, on-frequency or not. Additional anthropometric data for each participant is available in Appendix D.

**Table 2 biomedicines-11-01813-t002:** Pain thresholds in microamps.

	FPL	FDS	EDM	ECRL	Biceps	UN
P1 (f)	-	1600 *	-	-	-	2200
P2 (f)	3100	2850	-	-	-	1600
P3 (f)	2100	-	3500	3625	1700	1750
P4 (m)	-	2900	-	-	-	-
P5 (f)	-	-	-	-	-	2100
P6 (m)	-	-	-	-	-	-
P7 (m)	-	-	-	-	-	1800
P8 (f)	-	-	3300	-	-	-
P9 (m)	-	-	-	3500	2300	-
P10 (m)	-	-	-	-	2500	3000
x̅:	2600	2450	3400	3562.5	2166.67	2075
σ:	707.11	736.55	141.42	88.39	416.33	505.72

They were recorded only after extending this study with 10 additional participants. The low sample size should be noted, and conclusions should be drawn accordingly. “*” indicates the one case where the pain threshold coincided with the motor threshold. The pain threshold was always higher than the motoric one.

**Table 3 biomedicines-11-01813-t003:** Experimental arrangement.

Duration (mins)	Session A	Session B
~40	Preparation	Preparation
4	EEG	EEG
5	TI (open eyes)	tACS (open eyes)
4	EEG	EEG
20	Resting	Resting
4	EEG	EEG
5	TI (closed eyes)	tACS (closed eyes)
4	EEG	EEG
20	Resting	Resting
4	EEG	EEG
5	Carrier (open eyes)	Sham (open eyes)
4	EEG	EEG
20	Resting	Resting
4	EEG	EEG
5	Carrier (closed eyes)	Sham (closed eyes)
4	EEG	EEG
Σ: 150		

**Table 4 biomedicines-11-01813-t004:** Average log10 change values per electrode sites of interest for all conditions.

		Conditions			
		CARRIER	SHAM	TI	TACS
Electrodes	Cp2	−0.8979	−1.1056	−0.7432	0.6253
	Oz	−0.9536	−0.5720	−0.9099	0.7336
	Cp1	−0.7179	−1.4043	−0.9849	0.1977
	P4	−1.0186	−0.7466	−0.9747	0.9252
	Pz	−0.9019	−1.1754	−0.9276	0.8633
	P3	−0.8121	−1.1348	−1.1260	0.6558
	x̅	−0.8837	−1.0231	−0.9444	0.6668

**Table 5 biomedicines-11-01813-t005:** The log10 change values and respective color codes (explained in the description for Figure 5) for all the 27 recording electrode sites of the 32 channel EEG recording cap.

	TI Condition			tACS Condition	
T8	0.272125489		P4	0.925223315	
FT10	0.22456835		Pz	0.863346774	
C4(i)	0.213728648		Oz	0.733634371	
F8	0.16996691		P3	0.655825775	
FC6	0.160410961		CP2	0.625270233	
Fp2	−0.226333074		TP9	0.259269595	
TP10	−0.229755255		CP1	0.197711138	
CP6	−0.249873489		TP10	0.194444075	
FT9	−0.297620048		P8	0.156708375	
F4	−0.299967229		O2(i)	0.149073753	
Fp1	−0.311295871		T7	0.143121651	
FC2	−0.314082038		CP6	0.125180447	
T7	−0.380909655		CP5	0.117631221	
F7	−0.554834372		FC2	0.117244358	
C3(i)	−0.56316899		O1(i)	0.114452437	
FC5	−0.587040624		C4(i)	0.075227576	
TP9	−0.600919494		FT9	0.069544589	
FC1	−0.60498731		T8	0.060385997	
P8	−0.661651596		C3(i)	0.051579593	
CP5	−0.728324236		P7	0.005418679	
CP2	−0.743186706		FT10	−0.002998872	
F3	−0.858435449		F7	−0.056168677	
Fz	−0.875789837		FC5	−0.060500885	
Oz	−0.909949054		Fz	−0.080730221	
O2(i)	−0.918284876		F4	−0.117813612	
P7	−0.922544338		FC6	−0.118546085	
Pz	−0.92761255		F8	−0.140442036	
P4	−0.974683537		FC1	−0.171883412	
CP1	−0.984877384		Fp1	−0.227682946	
O1(i)	−0.991932369		F3	−0.242240967	
P3	−1.125984125		Fp2	−0.277242507	

## Data Availability

Comprehensive data tables are available upon reasonable request.

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
