# Peer review of "Why Temporal Inference Stimulation May Fail in the Human Brain: A Pilot Research Study"

_biomedicines, 2023, doi:10.3390/biomedicines11071813_

Round 1
Reviewer 1 Report
The authors present a study of interferential stimulation from peripheral and CNS stimulation. My major comment is that interferential stimulation is not demonstrated in peripheral stimulation as this does not show that the fundamental characteristics of interferential stimulation drive the stimulus. On the other hand, kilohertz stimulation can also generate stimulation such as Russian or pre-modulated stimulation.
In the case of interferential stimulation for muscles stimulation:
a) There is no demonstration of steerability and depth.
b) There is no control in Table 1. I would expect Russian stimulation and low frequency (few Herz) at least to determine any difference from interferential stimulation
Based on these two points, the electrical stimulation results do not prove the effects of interferential stimulation. It is known that Russian or premodulated stimulation can, and the important point to clarify is that interferential stimulation characteristics are engaging stimulation.
Therefore, statements similar to the following are not based on the results.
Abstract: “We have found strong evidence for peripheral stimulation efficacy, but no effect of TIS on the central nervous system (CNS).”
Conclusion: “We have found evidence for TIS utility in the PNS, but not in the CNS.”
1. In Lines 717-721, how did ultrasound identify muscles? Please include the ultrasound images and examples of motion and isolated and not isolated movement.
2. Regarding the statement, “Now we used ultrasound once again to determine the depth of the muscle movements (in millimeters) and to determine how isolated the muscle movements were on a subjective scale of 1 to 10, where 10 meant perfect isolation and 1 meant that all surrounding muscles were moving too (the rating was done by the same person in each two sessions to increase statistical reliability).”
No report of this subjective scale results in the appendix or main text.
This is necessary to determine if there was any deep muscle stimulation by interferential stimulation. More important is to show the ultrasound images demonstrating the effects of interferential stimulation in deep muscles.
3. In experiment 3
Why were different injection currents used between the four stimulation conditions? Moreover, what are the stimulation target differences between the selected montages?
6) Please explain which results show this statement
“Successful focal modulation in the depth of a well-defined, low-frequency electric field envelope in the muscles of the upper limb was found to be a reliable technique.”
7). Please make the following clarifications:
a) Please distinguish between the two electrode pairs in Figure 1.
b) Also, please include illustrative locations for each muscle locations.
c) Is the same intensity change for two channels in Table 1?
d) What are the parameters of TI for experiment 1
e) Did you get ethical permissions?
f) How is the ACS applied in Figure 3?
g) Line 321, check the enumeration of the experiment.
N.A
Author Response
/Attaching this whole reply also as a .docx/
Thank you very much for the detailed review. Upon your comments and suggestions, it became apparent that we should have included portions of our descriptions in the main text instead of where they ended up: the appendices. I will point out these edits at the relevant parts of the review.
Let me start by summarizing the key points that we base our conclusions on, they will be touched upon at later points:
P1) muscle movement proves the occurrence of temporal interference that can be used to innervate the PNS, proven by it being on the modulated frequency (see videos).
P2) Phosphene, a very easily inducible phenomenon that is hard to avoid even when unintended could not be triggered by the same interference
P3) Even though tACS did increase alpha power in the EEG study, the effect of TIS was on pair with the control conditions.
It is true that we could not prove steerability, practically usable intensities seem to be too low – at least in our design – to both shift the locus AND stay above contraction thresholds. This is in line with what Grossman and colleagues theorized about this phenomenon “it is possible to steer the envelope peak to have its maximum at essentially any depth throughout a volume, albeit with a tradeoff between the locus depth and its width and strength.”
As a sidenote, this is expected to be even more of a problem if muscles get activated in discreet leaps when innervated, from muscle belly to muscle belly rather than in a smooth and continuous fashion by muscle fiber recruitment, which seems to be the case. Net amplitudes would simply need to get too high for human dimensions.
Failure to show steering did not change the outcome of our key points, however: without this component, on-frequency muscle movement is still a clear sign of the occurrence of temporal interference, albeit without meaningful assumptions pointing to its depth relations. This latter matter is of relevance to the ultrasound videos you requested in one of your upcoming points Video S3-Video S4, and please also view the video about the on-frequency beating: Video S5-Video S7
Few other points:
Our current was continuous sinusoidal and we would be happy to find that it leads to similar outcomes than other established methods that for various reasons cannot be promoted into the CNS.
Correction: “We have found strong evidence for peripheral stimulation efficacy, but no effect of TIS on the central nervous system (CNS).” <- We have found strong evidence for stimulation efficacy on the modulated frequency in the PNS, but we found no evidence for its utility in the CNS. – line 17
An alternative statement could be the following: “We have found evidence for TIS as a continuous sinusoidal alternative to established stimulation waveforms in the PNS, but no evidence for its utility in the CNS.”
We have mentioned (although indeed have not emphasized in the table) that we did control stimulation with low Hz ACS and carrier control conditions, but they did not result in muscle movement at any time. We have now extended the table description with the following statement:
“Neither the carrier nor the ACS control conditions produced any movements, on-frequency or not.” Line 227, table description
The reference to these control conditions in the appendix that I now moved into the main text from line 193
“After rating, we have decreased the stimulation delta frequency to 0 Hz (carrier control condition) and held it for five seconds, looking for movements. After the five seconds, we have set the Δ frequency to 1 Hz from which we have slowly increased it up to the point where the rhythm of muscle movements could not be distinguished into discrete beats anymore and only a continuous, tonic contraction that lasted for more than a second could be observed. The participants then were asked if the stimulation was less comfortable, more comfortable or resulted in the same level of (dis)comfort whenever the stimulation Δ frequency was at either end of the applied frequency range. Finally, in 10 of the 20 participants, we have used the acquired pain threshold minus 50 µA to perform conventional ACS (only one active electrode pair delivering currents with 5 Hz, slowly ramped up to 10 Hz) for a final five second as a control condition (ACS control) during which we looked for detectable muscle movements again.”
Having said all of this, if you have any suggestions, for example changing the title of the work to reflect these better, please let us know.
To your other suggestions:
1) In Lines 717-721, how did ultrasound identify muscles? Please include the ultrasound images and examples of motion and isolated and not isolated movement.
From line 162: “To locate the correct position of the targeted muscle, we asked the participant to execute its function (e.g. repeatedly flex the middle finger for finding the FDS) whilst identifying the muscle via ultrasound.”
Please see attached Video S1, Video S2
2) Regarding the statement, “Now we used ultrasound once again to determine the depth of the muscle movements (in millimeters) and to determine how isolated the muscle movements were on a subjective scale of 1 to 10, where 10 meant perfect isolation and 1 meant that all surrounding muscles were moving too (the rating was done by the same person in each two sessions to increase statistical reliability).”
No report of this subjective scale results in the appendix or main text.
This is necessary to determine if there was any deep muscle stimulation by interferential stimulation. More important is to show the ultrasound images demonstrating the effects of interferential stimulation in deep muscles.
We have initially scrapped reporting this data but your call for its necessity is convincing:
(Focus) in subjective score* and depth in millimeters for 10 participants
|
|
FPL |
FDS |
EDM |
ECRL |
Biceps |
UN |
|
P1 (f) |
- |
15 |
- |
- |
- |
- |
|
P2 (f) |
(6) 5-10 |
(6) 7-15 |
- |
(3.75) 5-12 |
Everything |
(0) |
|
P3 (f) |
- |
(3) 6-16 |
(6.3) 7.8-16 |
- |
(8) 6-35 |
- |
|
P4 (m) |
- |
(4) 8-17 |
- |
(5.6) 5-16 |
- |
(1) 8-35 |
|
P5 (f) |
(7.7) 8-17 |
(10) 7-16 |
- |
(4.4) 7-15.5 |
(3) 8-35 |
(2) 2.7-3.5 |
|
P6 (m) |
(1) 3-35 |
(3) 4-17 |
(1.5) 9.5-17.5 |
- |
Everything |
- |
|
P7 (m) |
(3) 4-4.4 |
(3.7) 3-18 |
(2) 5-35 |
(1,4) 2.5-15 |
- |
(3) 8-19 |
|
P8 (f) |
- |
- |
- |
- |
-- |
- |
|
P9 (m) |
(2.5) 4-16 |
- |
(2.2) 7-22 |
(1) 4-16 |
(0) 6-35 |
- |
|
P10 (m) |
(8.8) 10.5-18 |
(5) 10-14 |
(7.6) 4-15 |
(1) 3-35 |
(1) 2-35 |
(3.8) 7-12 |
*the larger the score the more focused the contraction
**Everything means the whole diameter of the biceps was moving
On ultrasound the contractions definitely happened at depth, but it is not clear whether it was due to the modulated signal travelling from superficial areas or if it was generated at depth in the first place. In our impression, the nearest nerve to the source that is leading to the muscle belly gets innervated first which explains the leaps.
The general trend was limited focus of contractions in the forearm, and very diffuse twitching in case of the biceps.
3) In experiment 3
Why were different injection currents used between the four stimulation conditions? Moreover, what are the stimulation target differences between the selected montages?
The montages are detailed from line 519:
TIS, carrier control & sham control: O1-C3 and O2-C4
tACS: O1-C4 and O2-C3
Both montages target the occipital cortex at large, for the interferential conditions, the same montage was used as in the original Grossman paper that seems to be the standard in other TI phantom/simulation studies as well, a wider setup to achieve deeper envelope injection. For the traditional tACS, even though it wasn’t essential to use a traditional tACS style, it wasn’t necessary to keep the interferential setup either, but we seemed to only gain focus without any apparent loss so decided in favor of it.
Pasted from line 533:
For the interferential conditions we decided in favor of a wider montage, mirroring the setup of Grossman et al. [1], but for tACS we kept a more traditional one for better focus seemingly without losing anything in return.
4) Please explain which results show this statement
“Successful focal modulation in the depth of a well-defined, low-frequency electric field envelope in the muscles of the upper limb was found to be a reliable technique.”
This was shown by the on-frequency movement of the muscles during TIS, we hope the new videos demonstrate it well. We have referred to the movement being on-frequency on multiple occasions in the manuscript but now we also uploaded the videos with reference to this effect included. It is possible that the exact frequency deviated slightly from the input parameter to a small degree based on the original paper of Grossman and colleagues but the trend was continuous: “We validated TI stimulation on a population of cortical cells (Figure 1Ii) and found that interferential stimulation with a difference frequency of 10 Hz resulted in spike frequencies of 10.21 ± 0.83 Hz (mean ± SD), for a 1 kHz carrier frequency (n = 6 cells from 2 mice) and 9.68 ± 0.85 Hz for a 2 kHz carrier frequency”
5) Please make the following clarifications:
- a) Please distinguish between the two electrode pairs in Figure 1.
Drawn lines, wrote: Ep1: first electrode pair; Ep2: second electrode pair, line 171
- b) Also, please include illustrative locations for each muscle locations.
Do you mean here a picture that is showing the muscles of the forearm that we targeted? If so, this is the only point where we would respectfully ask for the opinion of the Editor on the matter while fully appreciating your suggestion.
- c) Is the same intensity change for two channels in Table 1?
Please let me know if I misunderstand your question, we interpret it like this: “does the intensity change equals between the two channels during the intensity increases?” We have changed the text from line 177 to address this better:
Participants were asked to indicate if significant discomfort has been reached while increasing intensity in both channels simultaneously by the same amount.
- d) What are the parameters of TI for experiment 1
It’s in the beginning of Appendix C – finding motor thresholds:
“Stimulation was started at a modulated frequency of 5 Hz at 500 µA intensity that was increased slowly (~1.5 second intervals in the beginning, ~2.5 at the end) by 50 µA increments.”
The whole Appendix C as part of the Methods section for Experiment 1 is reattached., thank you (line 179)
- e) Did you get ethical permissions?
Yes, it is under the ‘Institutional Review Board Statement’ as asked by Biomedicines, from line 697: approved by the Ethics Committee of the University of Medical Center Göttingen, Germany (protocol code UMG 38/2/21, for Experiment 1., and protocol code UMG 35/3/21, for Experiment 2 and 3).
- f) How is the ACS applied in Figure 3?
Copied this info from the description of control stimulation to be clarified here too, thank you:
“In the control stimulation, only the two lower electrodes were turned on.”, line 382.
- g) Line 321, check the enumeration of the experiment.
Corrected, thank you for pointing this out.
Wehope the corrections and the new videos address most of the questions. If due to misunderstanding on our side they did not, pointing out particular deficits if remained would be very much appreciated. Thank you very much again for your careful review!

Reviewer 2 Report
In this study, Iszak et al. presented a comprehensive study to validate the effect of temporal interference stimulation (TIS) on human being healthy participants. Three experiments including muscle twitches, retinal phosphenes, and EEG alterations were conducted with 24 participants in total. The results of the muscle twitches experiment did not show selective muscle activation with TIS. The retinal phosphenes experiment showed only ACS control induced phosphenes but not TIS nor carrier control condition. Lastly, the EEG experiment showed that TACS increased while carrier, sham, and TI conditions all decreased alpha power. Based on these results, the authors argued the TIS may fail in the human brain (CNS). I think this study provided useful information, on a newly developed non-invasive brain stimulation method, although mostly negative results. However, there are several points not clear or convincing to me.
1. The rational of the assumption for experiment 1 is not very clear to me. The TIS can target at specific location with specific frequency, it was not clear here how the targeted location was controlled, and why we should expect to see the move of the focus point while lowering the electrode current placed on the medial side of the forearm. Could it make it vertically deeper rather than exteriorly lower?
2. Why there were so many missing values in Table 2? The information provided by this table was not clear, and any info provided would be questionable since there were more than half missing points for each participants.
3. In experiment 2, it was not very clear why should we expect to see phosphenes in TI and carrier stimulation conditions, since in carrier stimulation condition, the effect from two sources could somehow be cancelled, while in TI condition, the lack of effect could be due to the shifted focused site. Similar question for the exploratory experiment targeting visual cortex from the occipital scalp region. The authors reported phosphenes only in the ACS control condition, which could also be due to the inaccurate of the targeting site and/or the insufficient intensity.
4. In experiment 3, the authors “hypothesized that stimulation would cause a change in the power spectrum of occipital brain oscillations in both the TACS and the TIS conditions, but not in the CARRIER and SHAM conditions.” The results summarized in Table 4 showed power decrease in CARRIER, SHAM, and TI conditions, and increase in TACS condition, which somehow supported the hypothesis. First, if the decrease is significant, then the effect of TI was approved rather than questioned. Second, it could be possible that the decrease in alpha power was the phasic effect, i.e., TI happened to decrease rather than increase alpha.
5. Finally, there are inconsistency between the presented results, some of the conclusions, and the title. To me, at least the presented results did not really suggest the failure of TIS.
Minor points
“Experiment 1. – phosphene study” should be “Experiment 2. – phosphene study”
Author Response
/Attaching this whole reply in a .docx document/
In this study, Iszak et al. presented a comprehensive study to validate the effect of temporal interference stimulation (TIS) on human being healthy participants. Three experiments including muscle twitches, retinal phosphenes, and EEG alterations were conducted with 24 participants in total. The results of the muscle twitches experiment did not show selective muscle activation with TIS. The retinal phosphenes experiment showed only ACS control induced phosphenes but not TIS nor carrier control condition. Lastly, the EEG experiment showed that TACS increased while carrier, sham, and TI conditions all decreased alpha power. Based on these results, the authors argued the TIS may fail in the human brain (CNS). I think this study provided useful information, on a newly developed non-invasive brain stimulation method, although mostly negative results. However, there are several points not clear or convincing to me.
- The rational of the assumption for experiment 1 is not very clear to me. The TIS can target at specific location with specific frequency, it was not clear here how the targeted location was controlled, and why we should expect to see the move of the focus point while lowering the electrode current placed on the medial side of the forearm. Could it make it vertically deeper rather than exteriorly lower?
Target location was determined with ultrasound, please see from line 162 (we have also provided video of this locating on request, please see attached Video S1, Video S2).
“Live steering” is referenced at line 55 from Grossman et al. 2017 but this particular question concerning dimensions is not addressed in that work either and our study meant to test only the practical outcomes without validating the modelling.
”We found that by changing the current ratio between the electrode pairs, while keeping the current sum fixed, the peak envelope modulation became increasingly close to the electrode pair with the lower current, with the peak moving 20% of the radius away from the center in the 1:2.5 case (Figure 2D) and 35% of the radius away from the center in the 1:4 case (Figure 2E). This suggests the possibility of “live steering” of activity from one deep site to another within the brain, without having to physically move the electrodes themselves.”
We have just followed their setup of changing the current ratios, but did not wish to validate the concept mechanistically.
- Why there were so many missing values in Table 2? The information provided by this table was not clear, and any info provided would be questionable since there were more than half missing points for each participants.
Our apologies, we did not update the title of this table after splitting its content into another one, so it featured a reference to contractions still. Thank you you for pointing out the wrong title (line 230): Thresholds for contractions and (pain) in microamps. -> Pain thresholds in microamps
So as this table features pain thresholds only, and only about half of our participants experienced any such sensation during stimulation, we could have emitted its inclusion, but we thought that the lack of pain is informative in itself enough to keep. We meant to emphasize caution from line 231 too just like you: “The low sample size should be noted, and conclusions should be drawn accordingly”
- In experiment 2, it was not very clear why should we expect to see phosphenes in TI and carrier stimulation conditions, since in carrier stimulation condition, the effect from two sources could somehow be cancelled, while in TI condition, the lack of effect could be due to the shifted focused site. Similar question for the exploratory experiment targeting visual cortex from the occipital scalp region. The authors reported phosphenes only in the ACS control condition, which could also be due to the inaccurate of the targeting site and/or the insufficient intensity.
Shifted focus site might look concerning, but in practice phosphenes can be induced by stimulating at nearly every point above the neckline due to (probably) the current running on the surface of the head, reaching the eye, so modulated current in just the general proximity of the retina should induce phosphenes. Occipital stimulation for example is theorized to induce phosphenes due to currents travelling on the skin surface to the retina instead of direct occipital activation.
"Phosphene induction is a very robust and reliable marker of successful nerve stimulation with reasonably uniform between-subjects outcome, so much so, that inducting it unintentionally can be a major concern during the design of ACS protocols concerning cognitive performance (Schutter, 2016). It is also possible to induce phosphenes with transcranial magnetic stimulation (TMS) aimed at the visual cortex directly (Schaeffner & Welchman, 2017), albeit the resulting perception is a single flicker instead of a sustained visage as with ACS. Due to this reliability, we feel that reporting some observations drawn from our preliminary experiments (on one, two, or three individuals) can still be meaningful."
Also, the stimulation intensity being underpowered while possible, it seems highly unlikely due to the following result:
“All participants experienced phosphenes during the ACS control stimulation to establish the general sensation at the lowest used current strength of 100 μA.”
Also attached a video 8 that demonstrates on-frequency twitches without inducing phosphenes
Please note that carrier condition was not an assumption for phosphenes, only TI & ACS control (line 366).
Grossman, N., Bono, D., Dedic, N., Kodandaramaiah, S. B., Rudenko, A., Suk, H. J., Cassara, A. M., Neufeld, E., Kuster, N., Tsai, L. H., Pascual-Leone, A., & Boyden, E. S. (2017). Noninvasive Deep Brain Stimulation via Temporally Interfering Electric Fields. Cell, 169(6), 1029–1041. https://doi.org/10.1016/j.cell.2017.05.024
Schaeffner, L. F., & Welchman, A. E. (2017). Mapping the visual brain areas susceptible to phosphene induction through brain stimulation. Experimental brain research, 235(1), 205–217. https://doi.org/10.1007/s00221-016-4784-4
Schutter D. J. (2016). Cutaneous retinal activation and neural entrainment in transcranial alternating current stimulation: A systematic review. NeuroImage, 140, 83–88. https://doi.org/10.1016/j.neuroimage.2015.09.067Activation for the
- In experiment 3, the authors “hypothesized that stimulation would cause a change in the power spectrum of occipital brain oscillations in both the TACS and the TIS conditions, but not in the CARRIER and SHAM conditions.” The results summarized in Table 4 showed power decrease in CARRIER, SHAM, and TI conditions, and increase in TACS condition, which somehow supported the hypothesis. First, if the decrease is significant, then the effect of TI was approved rather than questioned. Second, it could be possible that the decrease in alpha power was the phasic effect, i.e., TI happened to decrease rather than increase alpha.
Here the active conditions were compared to the control ones and so the change was significant between only tACS and controls, but not between TIS and controls, so the results did not support the hypothesis. The effects for TIS were not significant as shown on line 574, and so the power decrease was unlikely due to intervention, the control stimulations leading to similar results.
I have rewritten the discussion from line 590 in the following way to reflect this better:
tACS leading to elevated alpha powers is in line with the results of previous studies, and our enhancement rates are most similar too [34]. TIS has failed to increase the EEG power, which could partially be attributed to the low sample size, however, the TACS condition did lead to significant alpha power changes at a sample size which was similar to or lower than what was typically used in previous studies [34,40]. Also, the values of the other conditions not only did not increase, but they have shifted into the opposite direction, TIS following a similar effect size than the control conditions, which means that the decrease was unlikely due to intervention or that both the TIS and the control conditions had a phasic effect. The values of both the CARRIER control and TIS could speak towards the set in of CB, but the negative shift was even more pronounced for SHAM. Values decreased throughout the whole cortex so if we still suspect the onset of CB, its effect should be considered immense and consequential.
Rewording:
TIS was not successful in increasing the EEG power -> TIS has failed to increase the EEG power line 591
what was typically used in previous -> what was typically used in previous studies line 594
We are excited to hear further suggestions if any comes to your mind. Thank you!
- Finally, there are inconsistency between the presented results, some of the conclusions, and the title. To me, at least the presented results did not really suggest the failure of TIS.
We base our conclusions on:
P1) muscle movement proves the occurrence of temporal interference that can be used to innervate the PNS, proven by it being on the modulated frequency (see videos).
P2) Phosphene, a very easily inducible phenomenon that is hard to avoid even when unintended could not be triggered by the same interference
P3) Even though tACS did increase alpha power in the EEG study, the effect of TIS was on pair with the control conditions.
We hope these corrections in the text did clarify some points for all readers. If due to misunderstanding on our part the provided points / videos did not address your core questions, pointing out particular deficits would be much appreciated.
Minor points
“Experiment 1. – phosphene study” should be “Experiment 2. – phosphene study”
Corrected, thank you!
Krisztián Iszak

Reviewer 3 Report
The study by Iszak et al. aims at evaluating the effectiveness of electrical stimulation based on temporal interference for the activation of peripheral nerves, retinal ganglion cells and cortical brain regions. The idea is interesting and the study is well presented. Unfortunately, most of the results (PNS steering, CNS and retinal stimulation) are negative, which should not be a compromising aspect for an experimental investigations based on established methods, but in the case of a methodological validation actually strongly limits the impact and interest of the evidence. Indeed, most failures might be due to technical choices rather than physical limitations of this approach.
Beside this general comment, which should be taken into account in the title and in the discussion more explicitly, I think that (major points):
1) the positive result obtained with PNS still requires some further, less anecdotal, evidence. EMG or Videos, as well as temporal analysis of the latter, should be provided to testify that muscle contraction is in phase with interference frequence.
2) ethical concerns relate to the apparently non designed experiments performed with phosphene stimulation, e.g. subjects modifying the electrode configurations and occipital stimulation. In particular, those high current densities (4 mA with the described electrode shapes) could have caused strong discomfort and manual placing by experimental subjects is warning.
3) better description of the custom modification of the stimulation devices as well as the testing results should be provided in the manuscript
Minor errors:
l 321 should be "Experiment 2"
I only detected some typos or unusual words
Author Response
/Attaching this whole reply as a .docx document also/
The study by Iszak et al. aims at evaluating the effectiveness of electrical stimulation based on temporal interference for the activation of peripheral nerves, retinal ganglion cells and cortical brain regions. The idea is interesting and the study is well presented. Unfortunately, most of the results (PNS steering, CNS and retinal stimulation) are negative, which should not be a compromising aspect for an experimental investigations based on established methods, but in the case of a methodological validation actually strongly limits the impact and interest of the evidence. Indeed, most failures might be due to technical choices rather than physical limitations of this approach.
Beside this general comment, which should be taken into account in the title and in the discussion more explicitly, I think that (major points):
1) the positive result obtained with PNS still requires some further, less anecdotal, evidence. EMG or Videos, as well as temporal analysis of the latter, should be provided to testify that muscle contraction is in phase with interference frequence.
We are now including a bundle of videos from various sessions, some of them aimed to demonstrate the contractions being on-frequency. As mentioned in the text, up to around 7 Hz, digit movements followed a very clear mirroring of the modulated frequency, from which point the tonic character of the movement started to cloud the assessment. Inserted a new part into the text from Line 254:
“On-frequency muscle contractions are demonstrated on Video S3-S7.” The videos are attached in the supplements.
2) ethical concerns relate to the apparently non designed experiments performed with phosphene stimulation, e.g. subjects modifying the electrode configurations and occipital stimulation. In particular, those high current densities (4 mA with the described electrode shapes) could have caused strong discomfort and manual placing by experimental subjects is warning.
All of our experiments were designed and ethical approval were got before we started the whole study. 4.1.2 / 1) of our ethical approval “UMG 35/3/25“
In our experience, manual placing with the electrodes not being attached has the opposite effect: it allows for the immediate cessation of noxious stimulus by the abrupt withdrawal of the electrode from the skin surface. Intensities were tuned upon the live feedback of the participant, based on their individual toleration.
3) better description of the custom modification of the stimulation devices as well as the testing results should be provided in the manuscript
I extended the description with the following customization parameters from line 90:
The custom modification is a broader bandwidth up to 2 kHz as follows:
- two advanced DC Stimulator Plus in Remote control connected to the signal generation unit and a resistor bridge at the output (-3 dB bandwidth > 3 kHz)
- cDAQ-9136 CompactDAQ controller (running LabVIEW)
- NI9260 dual channel voltage generation module providing remote control signals (24 bit, 51 ksps)
- NI USB-6255 Multifunction I/O Device with two analog input channels active. Input range: ± 10 V (16 bit 50 ksps)
Further descriptions of the customization parameters can be found in the work of Hunold, and Hunold and colleagues [8, 9/31].
Hunold, A., Berkes, S., Schellhorn, K., Antal, A., and Haueisen, J. (2020). Feasibility of new stimulator setup for temporal interference tES and its application in a homogeneous volume conductor. Poster. In International Conference on Non-Invasive Brain Stimulation (NIBS). https://doi.org/10.1016/j.clinph.2019.12.348
Hunold, A. (2021). Transcranial electric stimulation: modeling, application, verification. PhD Thesis. 95-96. https://doi.org/10.22032/dbt.49291
Also, a testing documentation of the equipment exists but not as an official public document now. Please let us know if this is sufficient explanation or if not, please tell us about what particularities / specific requirements you would like to learn more about. Thank you!
Minor errors:
l 321 should be "Experiment 2"
Thank you, corrected!

Round 2
Reviewer 1 Report
The authors have addressed my comments.
The addition of the videos is an important reference for interested readers. z
Reviewer 2 Report
Thanks for the response. I have no other comments.
Reviewer 3 Report
The authors replied to all raised issues.
No comment.